# Effects of Climate and Land Use changes on Vegetation Dynamics in the Yangtze River Delta, China Based on Abrupt Change Analysis

**Lei Wan** [1,2,3,4], **Huiyu Liu** [1,2,3,4,*], **Haibo Gong** [1,2,3,4] and **Yujia Ren** [1,2,3,4]

1   Jiangsu Center for Collaborative Innovation in Geographical Information Resource Development and Application, Nanjing Normal University, Nanjing 210023, China; 10150101@stu.njnu.edu.cn (L.W.); 161302008@stu.njnu.edu.cn (H.G.); 171302012@stu.njnu.edu.cn (Y.R.)
2   State Key Laboratory Cultivation Base of Geographical Environment Evolution (Jiangsu Province), Nanjing Normal University, Nanjing 210023, China
3   Key Laboratory of Virtual Geographic Environment (Nanjing Normal University), Ministry of Education, Nanjing Normal University, Nanjing 210023, China
4   College of Geography Science, Nanjing Normal University, Nanjing 210023, China
*   Correspondence: liuhuiyu@njnu.edu.cn; Tel.: +86-18951838599

**Abstract:** Vegetation dynamics is thought to be affected by climate and land use changes. However, how the effects vary after abrupt vegetation changes remains unclear. Based on the Mann-Kendall trend and abrupt change analysis, we monitored vegetation dynamics and its abrupt change in the Yangtze River delta during 1982–2016. With the correlation analysis, we revealed the relationship of vegetation dynamics with climate changes (temperature and precipitation) pixel-by-pixel and then with land use changes analysis we studied the effects of land use changes (unchanged or changed land use) on their relationship. Results showed that: (1) the Normalized Vegetation Index (NDVI) during growing season that is represented as GSN (growing season NDVI) showed an overall increasing trend and had an abrupt change in 2000. After then, the area percentages with decreasing GSN trend increased in cropland and built-up land, mainly located in the eastern, while those with increasing GSN trend increased in woodland and grassland, mainly located in the southern. Changed land use, except the land conversions from/to built-up land, is more favor for vegetation greening than unchanged land use (2) after abrupt change, the significant positive correlation between precipitation and GSN increased in all unchanged land use types, especially for woodland and grassland (natural land use) and changed land use except built-up land conversion. Meanwhile, the insignificant positive correlation between temperature and GSN increased in woodland, while decreased in the cropland and built-up land in the northwest (3) after abrupt change, precipitation became more important and favor, especially for natural land use. However, temperature became less important and favor for all land use types, especially for built-up land. This research indicates that abrupt change analysis will help to effectively monitor vegetation trend and to accurately assess the relationship of vegetation dynamics with climate and land use changes.

**Keywords:** climate change; land use change; NDVI; Yangtze River Delta; abrupt change

## 1. Introduction

Vegetation is an important component of terrestrial ecosystems, which plays a vital role in various ecological processes at different scales, reflecting the state of ecosystems [1–3]. The Normalized Vegetation Index (NDVI) is an important piece of information for describing the state of vegetation, which adequately express the growth state of surface vegetation [4]. There are many studies that use

NDVI to monitor vegetation dynamics, land degradation and ecosystems status at different scales [5–8]. Trend detection in NDVI time series can help to identify and quantify the changes in ecosystem properties from local to global scale [9]. Therefore, the study on vegetation dynamics can help us understand the structure and function of local ecosystems and assess the quality of ecosystems.

A large number of studies [10–12] have focused on the trend of vegetation dynamics, however, vegetation dynamics is non-linear and non-stationary [13–15]. Affected by extreme events such as extreme temperature, rainfall and drought, the slope of the gradual variation in vegetation may significantly change, which is defined as an abrupt change [8,16,17]. Vegetation trends may shift after abrupt changes such as from greening to browning or browning to greening. Meanwhile, the relationship between climate and vegetation may shift. Research on the abrupt change will allow us to monitor vegetation trends more accurately [13–15], while how the vegetation trends and the relationship with climate and land use changes varied remains unclear.

Vegetation dynamics is thought to be driven by both climate and land-use changes at regional, national and even global scales [18–20]. Temperature and precipitation are the main indicators used to describe climatic conditions, which can significantly affect vegetation growth [20,21]. In most dry areas, vegetation growth is positively correlated with precipitation, while there is a negative correlation between vegetation conditions and heavy rainfall in humid areas [22,23]. The increase in temperature is conducive to the vegetation growth in areas with strong water carrying capacity, while the vegetation growth in water-poor areas is severely restricted [24,25]. Thus, lack of study on the impacts of climate change may not correctly reveal the driving mechanisms of vegetation dynamics. Meanwhile, vegetation is highly determined by current land use. Land use changes lead to substantial variations in vegetation, they are considered to be the most important driving forces for vegetation dynamics [13]. Human activities such as urbanization, mining and reforestation have changed land use types greatly and further cause corresponding degradation or greening of vegetation [19,26,27]. Such as the project of returning cropland to forests and grasses play a positive role [28], while other activities such as expansion of urban land area mainly play a negative role, leading to vegetation browning [29]. The effects of climate on vegetation growth varied under different land use types [30,31]. Grassland is usually more sensitive to climate changes than forestland and cropland [32–34]. Some studies have revealed the effects of climate and land use changes on vegetation dynamics but were mostly focused on the trends of vegetation dynamics during different time periods [35–37]. It is still not clear how the relationship of vegetation with climate and land use changes varied after abrupt change. Abrupt change analysis will help deeply understand the effects of climate and land use changes on vegetation dynamics.

The Yangtze River Delta is an alluvial plain before the Yangtze River flows into the sea. It is an important part of the middle and lower reaches of the Yangtze River and plays an important role in China's economic field [38]. Monitoring vegetation dynamics will help to better understand the ecosystem quality and further to provide ecological support for economic development. In recent decades, companied with the urbanization process in the Yangtze River Delta, there are a continuous increase in temperature, an increase of variability in precipitation and a rapid land use changes [39,40]. The development of urban agglomerations has led to significant variations in the distribution of land use types in the Yangtze River Delta [41,42]. Therefore, it is of great significance to study the relationship of vegetation dynamics with climate and land use changes in the Yangtze River Delta but how the relationship varied after the abrupt changes remained unclear.

Thus, ignoring abrupt changes will not explore the hidden trend shifts from browning to greening/greening to browning and thus overestimate/underestimate the possible risk of vegetation degradation. To assess vegetation trends and the relationship with climate and land use changes after abrupt change can help us better monitor vegetation dynamics and maintain the sustainability of ecosystem in the Yangtze River Delta. In this study, the Mann-Kendall test [43,44] was used to detect the vegetation trends and its abrupt changes, correlation analysis and land use changes analysis were used to reveal the relationship of vegetation dynamics with climate and land use changes before

or after vegetation abrupt change. We processed the study pixel-by-pixel and thus it can reveal the spatial heterogeneity of the impact of climate and land use changes on vegetation dynamics. More specifically, the purpose of this study is to explore the following after abrupt changes happened—(1) the effects of land use changes on the variations of the spatial distribution of the vegetation trends in the Yangtze River Delta; (2) the variations of the relationship between GSN (growing season NDVI) and climatic changes (temperature and precipitation) pixel-by-pixel; (3) the effects of land use changes on the variations of the relationship between GSN and climatic changes. To this end, Sections 2.1 and 2.2 describes the study area and data processing. Sections 2.3–2.5 follow with a brief introduction of the methods. In Section 3.1, we studied the vegetation abrupt change and after that, how the effects of land use changes on vegetation trends varied were analyzed. In Section 3.2, the relationship between GSN and climate changes before and after abrupt change pixel-by-pixel and how land use changes affected the relationship were analyzed. Finally, we draw a discussion and conclusions in Sections 4 and 5.

## 2. Materials and Methods

### 2.1. Research Area

The study area is located in eastern China including Anhui province, Jiangsu province, Zhejiang province and Shanghai city (Figure 1), with an area of 348,000 square kilometers. This region is under a monsoon climate regime, with the regional average elevation of 140.17 m. The annual mean growing-season (from March to October) temperature is 18 °C–22 °C and the annual mean growing-season precipitation is 800–1400 mm. The two major river systems are the Yangtze River and Huaihe River. The Yangtze River Delta is one of China's most developed economic regions, with the Yangtze River Delta urban agglomeration, one of the six largest urban agglomerations in the world, which plays an important role in China's social and economic development. Meanwhile, the recent landscape change in the Yangtze River Delta is closely related to the process of urbanization and its land use type has varied significantly.

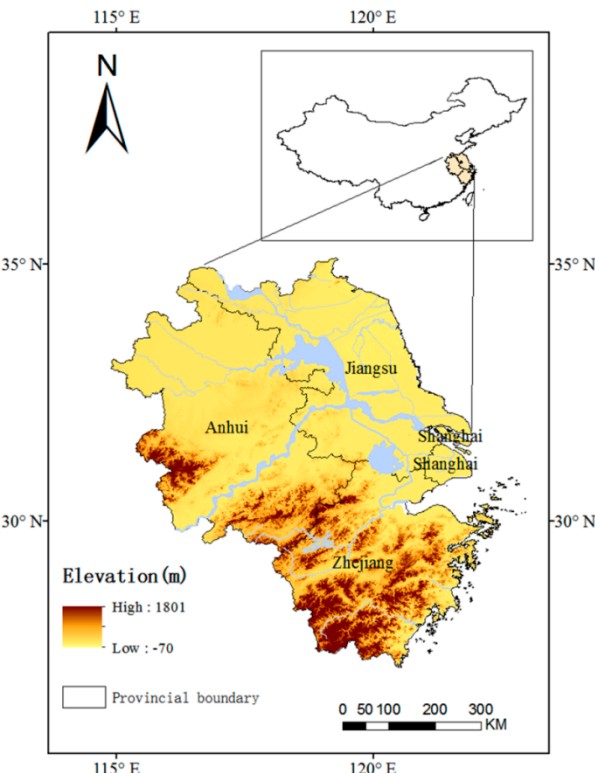

**Figure 1.** Map of the Yangtze River Delta.

*2.2. Data Processing*

In this study, the NDVI dataset was downloaded from the University of Arizona website (https://vip.arizona.edu) during 1982–2016 with a time resolution of 30 days and a spatial resolution of 5 km, which is one of the longest time series [45]. Then, all the climate and land use data were resampled to match the NDVI data with the nearest neighbor approach. Our NDVI data comes from the Vegetation Index and Phenology (VIP) Laboratory, which specializes in designing highly specialized data processing science algorithms and in generating consistent, high quality and well characterized Climate Data Record (CDR) and Earth Science Data Record (ESDR) long term single- and multi-satellite data products in support of global change studies. We used the average of March to October each year as the annual mean growing season NDVI (GSN) and then calculated the changes of GSN on each pixel.

For the convenience of correlation analysis with NDVI, climate data was chosen from 1982 to 2016 to match the time period of NDVI. Climate datasets (monthly temperature and precipitation data) for 58 meteorological stations in the study area were collected from the Chinese Meteorological Science Data Sharing Service Network (http://cdc.cma.gov.cn), China's official authority data center. In this study, the meteorological stations with missing data were eliminated. Based on the covariates of latitude, longitude and elevation (DEM) of each meteorological site (from the US Geological Survey), meteorological data (monthly temperature and precipitation data) were interpolated and resampled to continuous surface data with a 5 km spatial resolution to match the NDVI dataset, using thin-plate smoothing spline model in ANUSPLIN [46]. This interpolation method is affected by the number of sites and the accuracy of the DEM. Too few sample points participating in interpolation will lead to increased errors and poor interpolation accuracy. This method can not only interpolate the independent variables but also introduce a linear covariate submodel which provides the possibility of introducing multiple influence factors. This method is widely used in the climate interpolation field [47]. The growing-season (from March to October) mean precipitation and temperature were then considered as climatic factors.

Land use data sets in 1990 and 2015 were provided by Data Center for Resources and Environmental Sciences, Chinese Academy of Sciences (RESDC) (http://www.resdc.cn), with a spatial resolution of 1 km [48]. This data is official Chinese data, which is widely used in China [49]. Using 1st level classes in the Land use classification system in China (cropland, woodland, grassland, water body, built-up land and unused land) [49]. The economy is developed in the Yangtze River Delta and the area of unused land is small. We studied vegetation dynamics, so we did not consider unused land and water body but focused on the four main types of land use—cropland, woodland, grassland and built-up land (including urban land). Moreover, the land use data were resampled to 5 km spatial resolution to match the corresponding NDVI and climate data. To perform land use changes analysis, firstly, we set cropland, woodland, grassland, build-up land as 1, 2, 3 and 5 respectively and converted the property value of the land use data in 1990 and 2015 from text to number. Secondly, with the tool of Raster Calculator in ArcGIS10.2, we used land use data in 1990*10 plus the data in 2015 and then get the changes of land use from 1990 to 2015. For example, 12, 13, 15 means the conversion from cropland to woodland, grassland and build-up. Finally, we counted the area of each type of land use changes including changed and unchanged in the Attribute Table and put them into the table of transfer matrix. We used the land use change map to superimpose the vegetation trend map to obtain the impact of land use on vegetation dynamics. In addition, the maps of the correlation between climate and vegetation were superimposed with the land use change map to analyze the effects of different land use changes.

*2.3. Mann-Kendall Trend Test*

In this study, we adopt the Sequential version of Mann-Kendall trend test statistic [50] to detect the GSN trends and assess its significance during 1982–2016 in the Yangtze River Delta. The advantage of the Mann-Kendall trend test is that the sample does not need to follow a specific distribution and is

rarely disturbed by outliers and the calculation is simple. Mann-Kendall method has been widely used to assess the significance of trends [44,47].

Suppose there are time series of n sample quantities ($X_1$, $X_2$, ... $X_n$). For all k, j $\leq$ n and the distribution of k $\neq$ j, $X_k$ and $X_j$ are different, calculate the test statistic S, the formula is as follows:

$$S = \sum_{k=1}^{n-1} \sum_{j=k+1}^{n} Sgn\left(X_j - X_k\right). \tag{1}$$

Among them:

$$Sgn(X_j - X_k) = \begin{cases} +1 & \left(X_j - X_k\right) > 0 \\ 0 & \left(X_j - X_k\right) = 0 \\ -1 & \left(X_j - X_k\right) < 0 \end{cases}. \tag{2}$$

S is a normal distribution with a mean of 0 and a variance of $Var(S) = n(n-1)(2k+5)/18$. The statistical value Z is:

This is an example of an equation:

$$Z = \begin{cases} \dfrac{S-1}{\sqrt{Var(S)}} & S > 0 \\ 0 & S = 0 \\ \dfrac{S+1}{\sqrt{Var(S)}} & S < 0 \end{cases}. \tag{3}$$

According to the size of the statistical value Z, the trend of the sample with time series is observed. $Z > 0$ indicates that the sequence shows an increasing trend; on the contrary, it indicates a decreasing trend and it is judged whether the increase or decrease trend is significant within 95% of the confidence level. The Mann-Kendall trend test was done in MATLAB R2016b.

*2.4. Mann-Kendall Abrupt Change Test*

In this study, the Mann-Kendall nonparametric test was used to study the abrupt change in the GSN over the last 35 years in Yangtze River Delta region. This method is not only easy to calculate but also can find out the time of abrupt change. Therefore, it is a common abrupt change detection method and widely used in vegetation, hydrology and meteorology [51–53].

For a time series, X(t) = $x_i$, i = 1, 2, ...N, where $x_i$ is the independent and identically distributed and N is the number of data point. The statistic $d_k$ is defined as the following [54]:

$$d_k = \sum_{i=1}^{k} m_i, \ (2 \leq k \leq n), \tag{4}$$

with

$$m_i = \begin{cases} 1 & X_j - X_k > 0 \\ 0 & X_j - X_k < 0 \end{cases}, \ (j = 1, 2, \ldots, i). \tag{5}$$

In Equation $x_i$ and $x_j$ are the ith data value in time series.

Under the assumption that the original sequence is randomly independent, the mean and variance of $d_k$ are:

$$E(d_k) = k(k-1)/4 \tag{6}$$

$$Var(d_k) = k(k-1)(2k+5))/72. \tag{7}$$

The trends can be determined using the standard normal test statistic, $UF_k$ calculated from Equation (8):

$$UF_k = \frac{d_k - E(d_k)}{\sqrt{Var(d_k)}}, \ (k = 2, 3, 4, \ldots n). \tag{8}$$

$UF_k$ is a forward sequential statistic which is estimated using the original time series. For abrupt change point detection, the values of $UB_k$ can be computed backward starting from the end of the series. When the intersection point of the curves of $UF_k$ and $UB_k$ fell into the confidence interval, it is defined as the time of an abrupt change [55]. Then the significance was examined at the 99% level ($p < 0.01$). The Mann-Kendall abrupt change test was done in MATLAB R2016b.

*2.5. Pearson Correlation Analysis*

Studying the relationship between vegetation and its driving factors is mainly done by calculating and verifying correlation coefficients [56]. In this study, the relationship between GSN and climatic factors (temperature and precipitation) was analyzed. The formula is as the following:

$$R_{xy} = \frac{\sum_{i=1}^{n}(x_i - \bar{x})(y_i - \bar{y})}{\sqrt{\sum_{i=1}^{n}(x_i - \bar{x})^2 \sum_{i=1}^{n}(y_i - \bar{y})^2}}, \tag{9}$$

where n is the number of samples; $\bar{x}$ and $\bar{y}$ are the mean of the variables x and y, respectively; $R_{xy}$ is the correlation coefficient between the variables x and y. When R is greater than 0, it is a positive correlation and when R is less than 0, it is a negative correlation. Among them, it is significant at the 95% confidence level [57] and the other is insignificant.

In this study, based on the Mann-Kendall abrupt change detection method, the abrupt change of the GSN was analyzed. The Mann-Kendall trend analysis was used to calculate the spatial distribution of the GSN trends. The correlations between vegetation and climatic factors (temperature and precipitation) before or after abrupt change were analyzed using Pearson's correlation. The relationship between vegetation and climate change under different land use was obtained through superposition analysis.

## 3. Results

*3.1. Effects of Land Use changes on Vegetation Dynamics After Vegetation Abrupt Cchange*

### 3.1.1. Vegetation Dynamics after Abrupt Changes

In general, the growing season NDVI (GSN) value in the Yangtze River Delta showed an increasing trend (Figure 2). According to Mann-Kendall test statistics (Figure 3), it was found that the forward trend of annual GSN ($UF_k$) intersected the backward trend ($UB_k$) in the year 2000 within confidence interval. The intersection was considered as an abrupt change and we divided the entire study period into two periods—before (1982–1999) and after abrupt change (2000–2016). Before abrupt change, the GSN showed an insignificant trend. After then, the GSN trend shifted from insignificantly to significantly increasing.

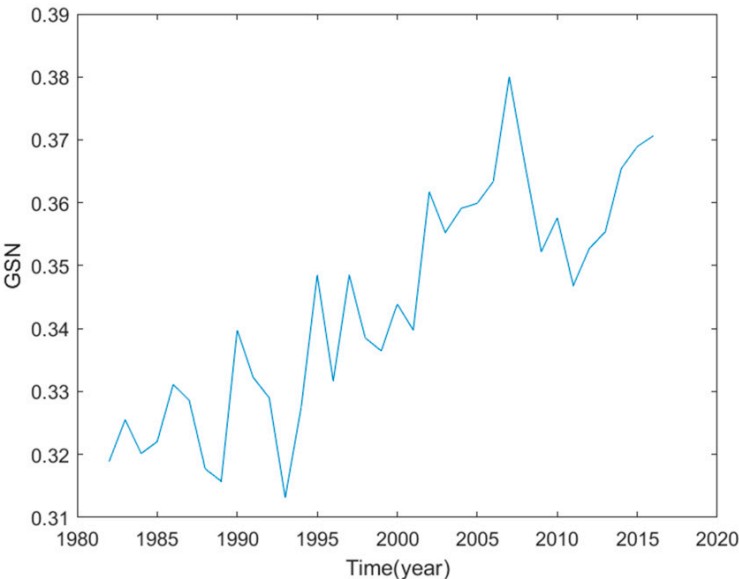

**Figure 2.** Annual mean GSN (growing season NDVI) changes in the Yangtze River Delta during 1982–2016.

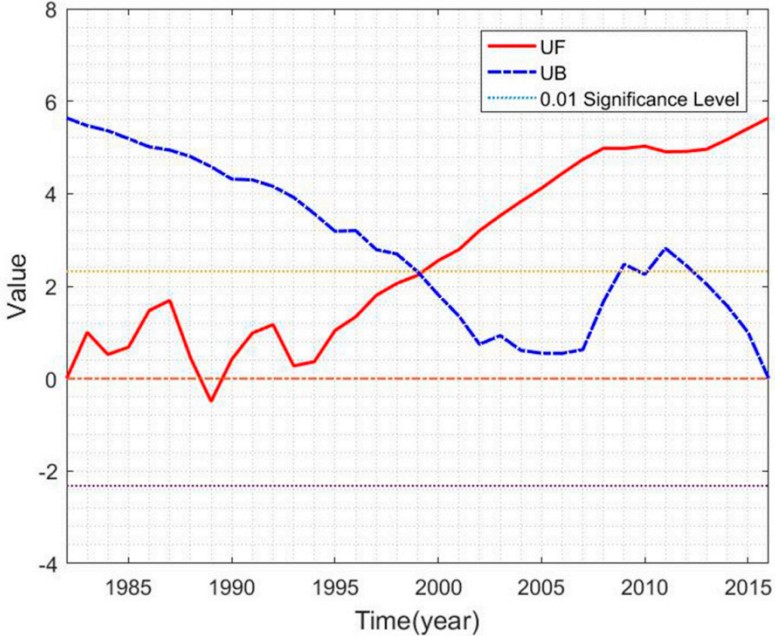

**Figure 3.** Abrupt change based on Mann-Kendall Abrupt Change Test.

In order to better reveal the variations in the vegetation trend, Mann-Kendall trend significance test was carried out for GSN during 1982–2016 and the two periods before and after abrupt change (Figure 4 and Table 1). It can be found that from 1982 to 2016, GSN was on the rise in 82.71% of the study area, among which 66.59% showed a significant trend, indicating that the vegetation in the whole study period mainly showed an upward trend. The significant decreasing area was located in the east of the Yangtze River Delta.

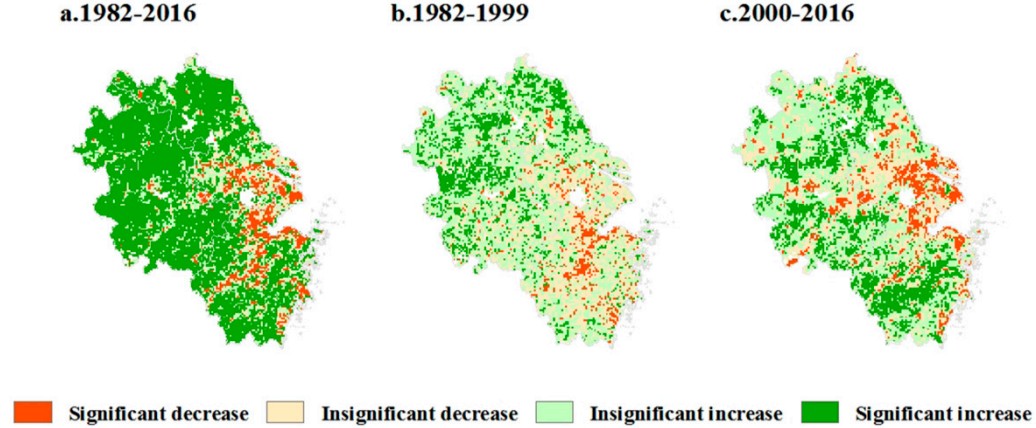

**Figure 4.** Trends of the GSN based on Mann-Kendall Trend Test during different periods (**a**) 1982–2016; (**b**) 1982–1999; (**c**) 2000–2016.

**Table 1.** Areas and percentages for different types of the trend of the GSN based on Mann-Kendall Trend Test in the Yangtze River Delta during different periods.

| Vegetation Trend | 1982–2016 | | 1982–1999 (Before Abrupt Change) | | 2000–2016 (After Abrupt Change) | |
|---|---|---|---|---|---|---|
| | Area/km$^2$ | Percentage/% | Area/km$^2$ | Percentage/% | Area/km$^2$ | Percentage/% |
| Significant decrease | 27,050 | 7.97 | 17,025 | 5.02 | 32,725 | 9.64 |
| Insignificant decrease | 31,625 | 9.32 | 94,225 | 27.78 | 76,300 | 22.48 |
| Insignificant increase | 54,725 | 16.12 | 164,275 | 48.42 | 148,800 | 43.85 |
| Significant increase | 226,000 | 66.59 | 63,700 | 18.78 | 81,550 | 24.03 |

The area of significant increase was small and mainly showed an insignificant increase either before or after abrupt change. Before abrupt change, in 67.2% of the study area, GSN showed an increasing trend and 18.78% showed significant increase (mainly in the north of the Yangtze River Delta) and 32.8% of the area suffered a certain degree of vegetation degradation, mainly distributed in the south of the Yangtze River Delta. However, after abrupt change, the vegetation trend shifted from insignificant decrease to significant increase in south of the Yangtze River Delta. The proportion showing a significant increasing trend increased to 24.03%. Meanwhile, the decreasing trends became more significant in the eastern regions of the Yangtze River Delta, accounting for 32.12% of the area.

3.1.2. Vegetation Trends under Different Land Use Changes

The map of land use changes (Figure 5) were obtained by overlaying the land use maps in 1990 and 2015 (Figure 5) and land-use conversion matrix was summarized in Table 2. The row data in Table 2 represent the land changes from 1990 to 2015. The total area in the last row corresponds to the area of each land-use type in 2015 and the total area in the last column corresponds to the area of each land-use type in 1990. The Yangtze River Delta is dominated by cropland and woodland. The cropland is mostly distributed in the northern part of the Yangtze River Delta and the woodland is mostly located in the south (Figure 5). The land use in the Yangtze River Delta has varied greatly from 1990 to 2015. The main types of land use changes included the conversions of cropland to built-up land, woodland to cropland, grassland to cropland and woodland and built-up land to cropland.

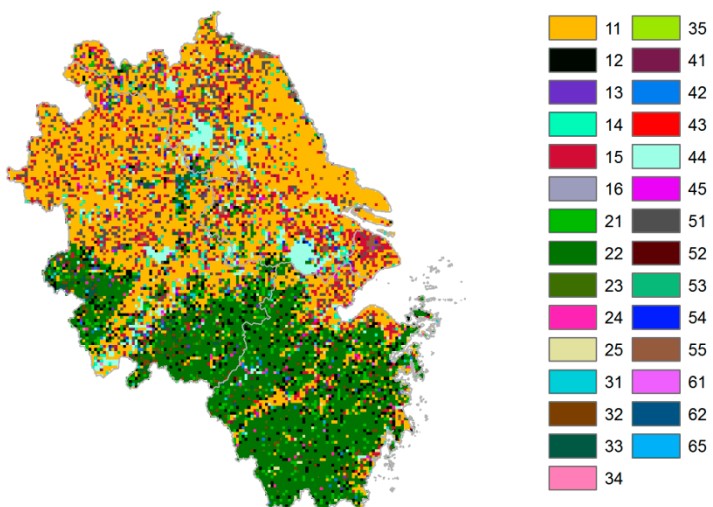

**Figure 5.** Use change in the Yangtze River Delta from 1990 to 2015 (1, cropland 2, woodland 3, grassland 4, water body, built-up land 6, unused land).

**Table 2.** Use change in the cropland, woodland, grassland and built-up land from 1990 to 2015.

| 2015 / 1990 | Cropland | Woodland | Grassland | Build-up Land | Sum |
|---|---|---|---|---|---|
| | Area/km$^2$ | Area/km$^2$ | Area/km$^2$ | Area/km$^2$ | Area/km$^2$ |
| Cropland | 134,925 | 15,150 | 1750 | 28,150 | 179,975 |
| Woodland | 14,800 | 79,725 | 4475 | 1975 | 100,975 |
| Grassland | 2475 | 3725 | 4225 | 200 | 10,625 |
| Built-up land | 17,600 | 700 | 175 | 7500 | 25,975 |
| Sum Area/km$^2$ | 169,800 | 99,300 | 10,625 | 37,825 | - |

The land use change and GSN trend map were superimposed to analyze the GSN variations under different types of unchanged and changed land use (Tables 3 and 4).

**Table 3.** Percentage of vegetation trends in different unchanging land use types in the Yangtze River Delta during different periods (%).

| Vegetation Trend | Cropland | | | Woodland | | | Grassland | | | Built-up Land | | |
|---|---|---|---|---|---|---|---|---|---|---|---|---|
| | TW | BA | AA | TW | BA | AA | TW | BA | AA | TW | BA | AA |
| Significant decrease | 8.80 | 3.63 | 12.14 | 2.76 | 5.26 | 1.68 | 0.63 | 2.53 | - | 14.23 | 6.05 | 17.08 |
| Insignificant decrease | 10.38 | 21.89 | 27.98 | 5.26 | 35.32 | 11.16 | - | 18.99 | 5.70 | 14.23 | 29.89 | 27.05 |
| Insignificant increase | 16.77 | 49.48 | 44.20 | 15.28 | 48.64 | 48.14 | 6.33 | 60.13 | 55.70 | 14.95 | 42.71 | 36.65 |
| Significant increase | 64.05 | 25.00 | 15.68 | 76.70 | 10.78 | 39.02 | 93.04 | 18.35 | 38.60 | 56.59 | 21.35 | 19.22 |

The whole period (WP), Before abrupt change (BA), After abrupt change (AA).

**Table 4.** Percentage of vegetation trends in different land use change types in the Yangtze River Delta during different periods (%).

| Type of Land Use Change | Decrease | | | | | | Increase | | | | | |
|---|---|---|---|---|---|---|---|---|---|---|---|---|
| | Significant | | | Insignificant | | | Insignificant | | | Significant | | |
| | TW | BA | AA | TW | BA | AA | TW | BA | AA | TW | BA | AA |
| Cropland–Woodland | 7.20 | 5.03 | 4.86 | 10.55 | 39.43 | 19.26 | 15.41 | 44.97 | 43.89 | 66.84 | 10.57 | 31.99 |
| Cropland–Grassland | 5.88 | - | 5.88 | 4.41 | 26.47 | 16.18 | 13.24 | 50.00 | 50.00 | 76.47 | 23.53 | 27.94 |
| Cropland–Built-up land | 18.47 | 7.39 | 21.62 | 13.96 | 24.32 | 29.91 | 16.49 | 49.37 | 35.86 | 51.08 | 18.92 | 12.61 |
| Woodland–Cropland | 6.73 | 6.39 | 4.49 | 9.50 | 34.03 | 17.27 | 16.41 | 47.32 | 45.25 | 67.36 | 12.26 | 32.99 |
| Woodland–Grassland | 3.41 | 3.98 | 3.41 | 1.70 | 25.57 | 7.95 | 12.50 | 52.84 | 43.75 | 82.39 | 17.61 | 44.89 |
| Woodland–Built-up land | 28.21 | 11.54 | 24.36 | 15.38 | 50.00 | 32.05 | 15.38 | 26.92 | 24.36 | 41.03 | 11.54 | 19.23 |
| Grassland–Cropland | 2.08 | 2.08 | 2.08 | 5.21 | 29.17 | 9.37 | 6.25 | 50.00 | 47.92 | 86.46 | 18.75 | 40.63 |
| Grassland–Woodland | 1.36 | 4.08 | 1.36 | 2.72 | 30.61 | 7.48 | 10.20 | 50.34 | 57.14 | 85.72 | 14.97 | 34.02 |
| Grassland–Built-up land | - | 14.29 | - | - | 14.29 | 71.43 | 57.14 | 28.57 | - | 42.86 | 42.85 | 28.57 |
| Built-up land–Cropland | 7.06 | 3.74 | 9.22 | 7.20 | 18.88 | 25.22 | 14.55 | 50.29 | 45.68 | 71.19 | 27.09 | 19.88 |
| Built-up land–Woodland | 15.38 | 11.54 | 3.85 | 23.08 | 50.00 | 34.62 | 19.23 | 34.62 | 46.15 | 42.31 | 3.84 | 15.38 |
| Built-up land–Grassland | 16.67 | 16.67 | 16.67 | 16.67 | 33.33 | 33.33 | 16.67 | 33.33 | 33.33 | 49.99 | 16.67 | 16.67 |

The whole period (WP), Before abrupt change (BA), After abrupt change (AA).

During the whole study period, GSN under the four land use types mainly showed a significant increase, especially for natural land use types as woodland and grassland (Table 3). The vegetation in the four land use types mainly showed an insignificant increase whether before or after abrupt change. After abrupt change, the decreasing trends in GSN in the cropland and built-up land increased, especially for significant decrease. The significant increase in GSN in the woodland and grassland increased greatly, while the insignificant decrease reduced obviously.

The vegetation was dominated by a significant increase under all types of land use changes during the whole period; however, the area percentages with a significant increase in the conversion from or to built-up land were much smaller than those from or to grassland (Table 4). After abrupt change, the significant decrease of GSN in the conversion of cropland and woodland to built-up land greatly increased, especially the conversion of cropland to urban areas. Moreover, the insignificant decrease of GSN in the conversion of grassland to built-up land greatly increased. In addition to the conversion to or from built-up land, the trends with significant increasing in the majority of changed land use types have increased. Meanwhile, except for the land use affected by urbanization, the proportion of significant increase in the changed land use was greater than that in unchanged land use.

*3.2. Effects of land use changes on Relationship between Annual Mean GSN and Climate Change*

To reveal the relationship between GSN and climate change before or after abrupt change, the correlation analysis method was used to calculate GSN, temperature and precipitation on the pixel scale at a 95% confidence level [57]. The insignificant correlation between precipitation and GSN in the Yangtze River Delta was the most from 1982 to 2016, which accounting for 96.38% (Table 5). 54.1% of the study area showed a negative correlation, mainly in the central and coastal areas of the Yangtze River Delta (Figure 6). 45.9% of the study areas were positively correlated and most were distributed in the north.

**Table 5.** Percentages of different correlation between precipitation and GSN in the Yangtze River Delta for different periods.

| Correlation | 1982–2016 | | 1982–1999(Before Abrupt Change) | | 2000–2016(After Abrupt Change) | |
|---|---|---|---|---|---|---|
| | Area/km$^2$ | Percentage/% | Area/km$^2$ | Percentage/% | Area/km$^2$ | Percentage/% |
| Significant negative | 9975 | 2.99 | 41,650 | 12.45 | 27,150 | 8.12 |
| Insignificant negative | 169,625 | 51.11 | 178,275 | 53.29 | 90,625 | 27.10 |
| Insignificant positive | 150,250 | 45.27 | 108,400 | 32.40 | 139,225 | 41.63 |
| Significant positive | 2075 | 0.63 | 6200 | 1.86 | 77,400 | 23.15 |

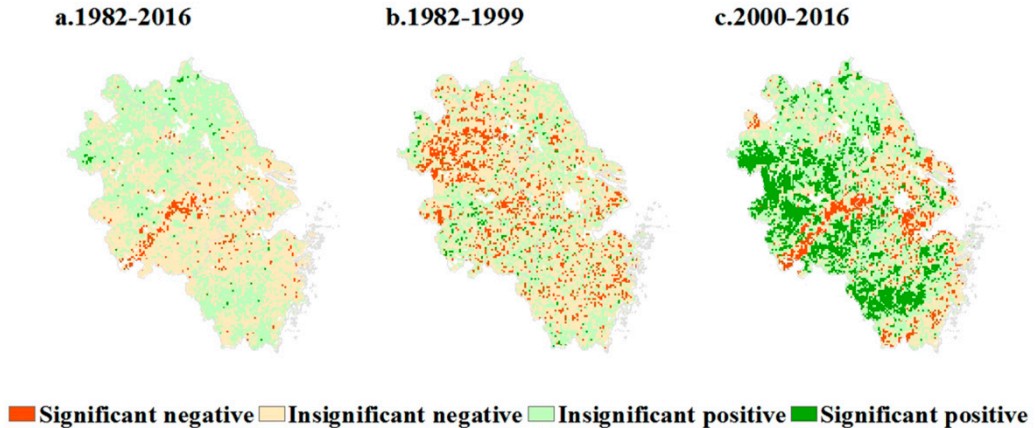

**Figure 6.** Between precipitation and GSN during different periods (**a**) 1982–2016; (**b**) 1982–1999; (**c**) 2000–2016.

Precipitation and GSN distribution patterns in the Yangtze River Delta have varied greatly after abrupt change. Precipitation was mainly negatively correlated with GSN (65.74%) before abrupt change. Among them, 12.45% showed a significant negative correlation and their distribution was relatively dispersed, mainly concentrated in northern Anhui Province. However, 64.78% of the study area showed a positive correlation with GSN after abrupt change, which was mostly from negative correlation in the west and south parts. Especially, significant positive correlation increased greatly from 1.86% to 23.15% mainly in the west, indicating that precipitation became more important for GSN. Meanwhile, the negative correlations decreased to 35.22%, concentrated in the east of the Yangtze River Delta.

Throughout the whole study period, for cropland, grassland and built-up land, the correlations between precipitation and GSN were mainly insignificant positive, while for woodland, the correlations was insignificant negative (Table 6). However, the correlation in the woodland was mainly shifted from the insignificant negative to the positive after abrupt change. Before abrupt change, the correlations in these four land use types were mainly insignificant, especially insignificant negative. While, the correlations shifted from negative to positive after abrupt change, especially significant positive increased greatly. In the meantime, the increase of significant positive correlations most occurred in woodland and grassland, while the increase of insignificant positive correlations most occurred in human-affected land use as cropland and built-up land.

**Table 6.** Percentage of different correlation between precipitation and GSN in different unchanging land use in the Yangtze River Delta during different periods (%).

| Correlation | Cropland | | | Woodland | | | Grassland | | | Built-Up Land | | |
|---|---|---|---|---|---|---|---|---|---|---|---|---|
| | TW | BA | AA | TW | BA | AA | TW | BA | AA | TW | BA | AA |
| Significant negative | 3.48 | 14.10 | 9.47 | 1.69 | 10.15 | 4.22 | 0.65 | 9.80 | 1.95 | 2.56 | 12.73 | 7.61 |
| Insignificant negative | 45.70 | 53.91 | 29.89 | 61.40 | 52.35 | 21.37 | 46.41 | 46.41 | 14.28 | 41.03 | 56.00 | 31.88 |
| Insignificant positive | 50.09 | 30.43 | 42.58 | 36.44 | 35.40 | 42.57 | 51.63 | 37.91 | 38.96 | 55.68 | 29.82 | 39.13 |
| Significant positive | 0.73 | 1.56 | 18.06 | 0.47 | 2.10 | 31.84 | 1.31 | 5.88 | 44.81 | 0.73 | 1.45 | 21.38 |

The whole period (WP), Before abrupt change (BA), After abrupt change (AA).

Precipitation and GSN were mainly insignificantly correlated under all types of land use changes throughout the whole study period (Table 7). Among them, correlations in the conversion from or to woodland were mainly insignificant negative, while in the conversion from or to land with high human influence were mainly insignificant positive. Before abrupt change, all types of land use change were dominated by insignificant negative correlations. After then, all types of land use changes mainly showed positive correlation and expect for the land use types affected by urbanization, others showed

significant positive correlation. Meanwhile, the proportion of significant positive correlations in most changed land was greater than unchanged land.

**Table 7.** Percentage of different correlation between precipitation and GSN in different land use change types in the Yangtze River Delta during different periods (%).

| Type of Land Use Change | Negative | | | | | | Positive | | | | | |
| | Significant | | | Insignificant | | | Insignificant | | | Significant | | |
| | TW | BA | AA | TW | BA | AA | TW | BA | AA | TW | BA | AA |
| --- | --- | --- | --- | --- | --- | --- | --- | --- | --- | --- | --- | --- |
| Cropland–Woodland | 7.20 | 9.43 | 5.37 | 10.55 | 52.52 | 26.17 | 15.41 | 35.69 | 32.72 | 66.84 | 2.36 | 35.74 |
| Cropland–Grassland | 5.88 | 19.41 | 4.48 | 4.41 | 49.25 | 19.40 | 13.24 | 31.34 | 52.24 | 76.47 | - | 23.88 |
| Cropland–Built-up land | 18.47 | 13.81 | 11.49 | 13.96 | 55.25 | 31.25 | 16.49 | 30.02 | 42.65 | 51.08 | 0.92 | 14.61 |
| Woodland–Cropland | 6.73 | 10.97 | 5.55 | 9.50 | 52.44 | 21.49 | 16.41 | 33.97 | 44.19 | 67.36 | 2.62 | 28.77 |
| Woodland–Grassland | 3.41 | 8.52 | 2.84 | 1.70 | 52.84 | 20.46 | 12.50 | 35.23 | 36.36 | 82.39 | 3.41 | 40.34 |
| Woodland–Built-up land | 28.21 | 3.95 | 11.69 | 15.38 | 55.26 | 41.56 | 15.38 | 39.47 | 20.78 | 41.03 | 1.32 | 25.97 |
| Grassland–Cropland | 2.08 | 14.58 | 2.08 | 5.21 | 50.00 | 16.67 | 6.25 | 32.29 | 37.50 | 86.46 | 3.13 | 43.75 |
| Grassland–Woodland | 1.36 | 9.52 | 2.04 | 2.72 | 55.11 | 14.96 | 10.20 | 34.69 | 42.18 | 85.72 | 0.68 | 40.82 |
| Grassland–Built-up land | - | - | 16.67 | - | 85.71 | 33.33 | 57.14 | 14.29 | 33.33 | 42.86 | - | 16.67 |
| Built-up land–Cropland | 7.06 | 13.31 | 6.13 | 7.20 | 52.97 | 27.30 | 14.55 | 31.40 | 50.51 | 71.19 | 2.32 | 16.06 |
| Built-up land–Woodland | 15.38 | 3.85 | 7.69 | 23.08 | 69.23 | 30.77 | 19.23 | 23.07 | 46.15 | 42.31 | 3.85 | 15.39 |
| Built-up land–Grassland | 16.67 | - | - | 16.67 | 50.00 | 50.00 | 16.67 | 50.00 | 33.33 | 49.99 | - | 16.67 |

The whole period (WP), Before abrupt change (BA), After abrupt change (AA).

From 1982 to 2016, 59.24% of the study areas showed a significant positive correlation between GSN and temperature (Figure 7, Table 8). The negative correlation was mainly distributed in the eastern part of the Yangtze River Delta, accounting for 13.16% of the study area. However, whether before or after abrupt change, the temperature was mainly insignificantly positive correlated with GSN. Before abrupt change, 77.6% of the area showed a positive correlation between temperature and GSN mainly distributed in most of the Yangtze River Delta except the east. Among them, only 24.31% showed significant increasing trend. After abrupt change, 63.84% of the study area was positively correlated with GSN and only 11.48% showed a significant positive. Moreover, the correlation showed a particularly obvious shift from positive to negative in the northwestern part of the Yangtze River Delta. The area with significant correlation reduced after abrupt change, indicating that the correlation between temperature and GSN became weaker and the influence of temperature on GSN was weakened.

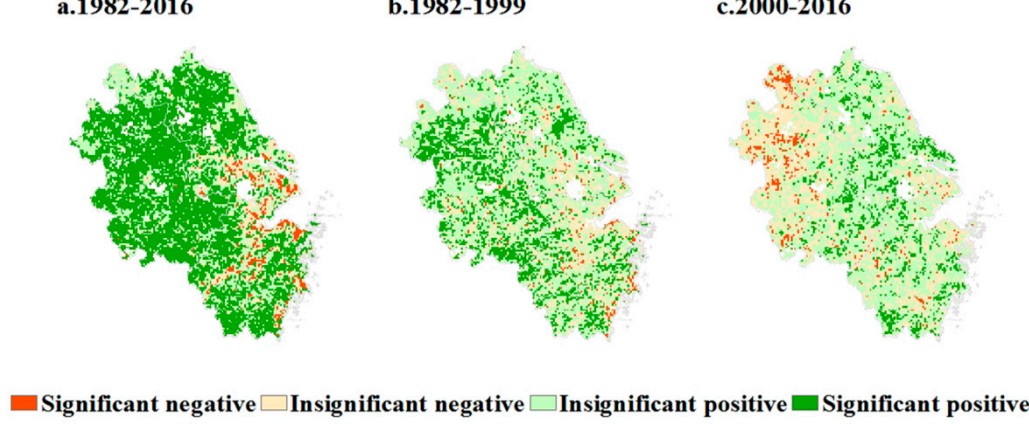

**Figure 7.** Between temperature and GSN during different periods (**a**) 1982–2016; (**b**) 1982–1999; (**c**) 20002016.

**Table 8.** Percentages of different correlation between temperature and GSN in the Yangtze River Delta during different periods.

| Correlation | 1982–2016 | | 1982–1999 (Before Abrupt Change) | | 2000–2016 (After Abrupt Change) | |
|---|---|---|---|---|---|---|
| | Area/km$^2$ | Percentage/% | Area/km$^2$ | Percentage/% | Area/km$^2$ | Percentage/% |
| Significant negative | 11,725 | 3.53 | 7100 | 2.12 | 12,700 | 3.80 |
| Insignificant negative | 31,950 | 9.63 | 67,850 | 20.28 | 108,225 | 32.36 |
| Insignificant positive | 91,600 | 27.60 | 178,275 | 53.29 | 175,100 | 52.36 |
| Significant positive | 196,600 | 59.24 | 81,300 | 24.31 | 38,375 | 11.48 |

During the whole study period, the correlations between temperature and GSN were significant in all the four land use types (Table 9), mainly significant positive. In the meantime, the area proportions of significant positive correlations in natural land use as woodland and grassland were much larger than those in human-affected land use as cropland and built-up land. Before abrupt change, the correlation under the four land use types was mainly positive, especially insignificant positive. After then, the significant positive correlation decreased greatly except built-up land but the insignificant correlation increased in all land use types. However, only for woodland, the insignificant positive correlation increased and for the other land use types, the insignificant negative correlations increased greatly.

**Table 9.** The percentage of different correlation between temperature and GSN in different unchanging land use in the Yangtze River Delta during different periods (%).

| Correlation | Cropland | | | Woodland | | | Grassland | | | Built-Up Land | | |
|---|---|---|---|---|---|---|---|---|---|---|---|---|
| | TW | BA | AA | TW | BA | AA | TW | BA | AA | TW | BA | AA |
| Significant negative | 3.63 | 1.62 | 4.46 | 1.24 | 2.16 | 2.25 | - | 1.31 | 3.25 | 6.23 | 1.82 | 2.90 |
| Insignificant negative | 10.05 | 17.48 | 34.15 | 6.20 | 21.09 | 28.96 | 0.65 | 27.45 | 45.45 | 16.12 | 26.18 | 38.41 |
| Insignificant positive | 28.75 | 56.62 | 50.29 | 25.21 | 48.70 | 56.31 | 27.45 | 47.71 | 46.75 | 31.50 | 56.36 | 45.65 |
| Significant positive | 57.57 | 24.28 | 11.10 | 67.35 | 28.05 | 12.48 | 71.90 | 23.53 | 4.55 | 46.15 | 15.64 | 13.04 |

The whole period (WP), Before abrupt change (BA), After abrupt change (AA).

Under the land use change throughout the whole study period, temperature and GSN were mainly significantly positive correlated (Table 10). However, the insignificant positive correlation was dominant whether before or after abrupt change. Moreover, after abrupt change, the insignificant negative increased greatly, while the significant positive was obviously reduced and the insignificant positive correlation in the conversion to built-up land was reduced. The proportion of significant positive correlations in most land-use changes was greater than the conversion to land use affected by urbanization.

**Table 10.** The percentage of different correlation between temperature and GSN in different land use change types in the Yangtze River Delta during different periods (%).

| Type of Land Use Change | Negative | | | | | | Positive | | | | | |
|---|---|---|---|---|---|---|---|---|---|---|---|---|
| | Significant | | | Insignificant | | | Insignificant | | | Significant | | |
| | TW | BA | AA | TW | BA | AA | TW | BA | AA | TW | BA | AA |
| Cropland–Woodland | 3.36 | 3.03 | 3.02 | 10.25 | 19.70 | 29.86 | 26.39 | 51.18 | 54.03 | 60.00 | 26.09 | 13.09 |
| Cropland–Grassland | 1.49 | - | 2.99 | 5.97 | 17.91 | 31.34 | 29.85 | 55.22 | 53.73 | 62.69 | 26.87 | 11.94 |
| Cropland–Built-up land | 9.50 | 3.41 | 5.60 | 17.80 | 22.10 | 36.40 | 26.10 | 53.31 | 49.36 | 46.60 | 21.18 | 8.64 |
| Woodland–Cropland | 3.66 | 2.61 | 3.99 | 8.73 | 24.39 | 27.21 | 28.62 | 48.78 | 55.28 | 58.99 | 24.22 | 13.52 |
| Woodland–Grassland | 3.41 | 2.27 | 1.71 | 1.14 | 17.61 | 28.41 | 21.02 | 52.84 | 59.09 | 74.43 | 27.28 | 10.79 |
| Woodland–Built-up land | 11.84 | 3.95 | - | 22.37 | 19.74 | 32.47 | 27.63 | 57.89 | 57.14 | 38.16 | 18.42 | 10.39 |
| Grassland–Cropland | 1.04 | - | 2.08 | 3.13 | 21.87 | 43.75 | 19.79 | 53.13 | 47.92 | 76.04 | 25.00 | 6.25 |
| Grassland–Woodland | 0.68 | 3.40 | 4.76 | 4.08 | 21.09 | 34.70 | 26.53 | 51.02 | 54.42 | 68.71 | 24.49 | 6.12 |
| Grassland–Built-up land | - | - | - | - | 14.29 | 28.57 | 28.57 | 57.14 | 42.86 | 71.43 | 28.57 | 28.57 |
| Built-up land–Cropland | 3.66 | 1.88 | 5.69 | 7.76 | 17.95 | 37.37 | 29.14 | 57.16 | 48.32 | 59.44 | 23.01 | 8.62 |
| Built-up land–Woodland | 11.53 | 7.69 | 11.54 | 23.08 | 34.62 | 30.77 | 23.08 | 50.00 | 38.46 | 42.31 | 7.69 | 19.23 |
| Built-up land–Grassland | 16.66 | - | - | - | 16.67 | 50.00 | 16.66 | 50.00 | 50.00 | 66.68 | 33.33 | - |

The whole period (WP), Before abrupt change (BA), After abrupt change (AA).

## 4. Discussion

*4.1. Effects of Land Use Changes on Vegetation Dynamicsin the Yangtze River Deltabased on Abrupt Change Analysis*

Our study found that the vegetation conditions in most areas of the Yangtze River Delta improved significantly during the period 1982–2016, which is consistent with other relevant research results [58,59]. In particular, our study found that the area showing a significant increasing trend in natural land use types such as woodland and grassland is much larger than that of human-affected land use types such as cropland and built-up land. The abrupt change in vegetation occurred in 2000, which was thought to be caused by climate factors and human activities [58,60]. Although vegetation trends were dominated by insignificant variations, either before or after abrupt change, the spatial distribution of the vegetation trend had shifted greatly. According to our research, it can be seen that after abrupt change, the increasing trend of vegetation in the woodland, mainly located in the south of the Yangtze River Delta increased greatly and the decreasing trend of built-up land and cropland in the eastern region increased obviously, which caused the variations of vegetation trends in the north and south. This indicated that the north-south vegetation trend of the Yangtze River Delta before or after abrupt change was closely related to the land use type. Some studies have suggested that the increase in the vegetation of woodland in the south of the Yangtze River Delta was related to human-related policies such as ecological protection projects [61]. The urbanization in the eastern cities and the irrational human use of cropland [62–64] had led to a great increase in vegetation browning of cropland and built-up land. For natural land use type as grassland and woodland, the increase in temperature after abrupt change may enhance photosynthesis and prolong the growing season, which is conducive to vegetation growth. Meanwhile, natural land such as grassland is less affected by urbanization on vegetation growth. Therefore, the increase of vegetation in the conversion from or to grassland is obviously greater than built-up land. After abrupt change, due to the development of urbanization, more vegetation showed browning trend in built-up land as well as the conversions to built-up land, especially the conversion of natural land such as grassland to built-up land. It can be seen that the conversions of cropland, woodland and grassland to built-up areas led to vegetation degradation and the negative impact of urbanization on vegetation in these areas was strengthened. It is found that the vegetation growth under changed land is better than that of the unchanged land, which is more conducive to vegetation growth. Through abrupt change analysis, vegetation trend shifted greatly. Therefore, only analyzing vegetation dynamics through the whole period is hardly to reveal the inherent conversion of vegetation dynamics and further to correctly monitor vegetation change, which may overestimate the risk of vegetation degradation in the southern and underestimate the risk in the eastern.

*4.2. Effects of Climate Changes on Vegetation Dynamics after Abrupt Change*

Precipitation and temperature are two important factors affecting vegetation growth [65,66]. Previous studies have found that the impact of precipitation on vegetation in wet areas is insignificant [67] and our study also found that the correlation between precipitation and GSN was insignificant during the whole period in the Yangtze River Delta. The Yangtze River Delta is rich in water system and abundant in precipitation and thus precipitation has few effects on vegetation [67]. After abrupt change, the correlation between precipitation and GSN became more significant and shifted from insignificant negative to positive, especially significant positive. Studies have suggested that temperature promotes vegetation growth in the region [67] and our research has also found that temperature was positively correlated with GSN throughout the whole study period, especially significantly positive correlated. At the same time, the negative correlation between temperature and GSN was mainly distributed in the eastern part of the Yangtze River Delta, where were dominated by urban land. Because temperature rises in urban will increase the surface evaporation, further reduce the surface water content and limit the growth of vegetation. However, after abrupt change,

the correlation between temperature and GSN shifted from significant positive to insignificant or even negative.

The temperature in the study area was on the rise during the whole study period and the increase after abrupt change was greater than before [68,69]. The increase in temperature promotes photosynthesis and prolongs the growing season, which contributes to the growth of vegetation. However, as the temperature rises further, the evaporation increases and the water stress of the vegetation increases. This makes not only the correlation between temperature and vegetation shift from positive to negative but also the correlation between vegetation and precipitation from significant negative to positive. After abrupt change, the significance between precipitation and vegetation increased greatly, while the significance between temperature and vegetation reduced obviously, indicating that precipitation had an increasing effect on GSN and the effect of temperature on GSN was weakened. So, abrupt change analysis is important for assessing the relationship between climate changes and GSN.

### 4.3. Impacts of Land use Changes on the Relationship between Climate Changes and GSN Based on Abrupt Change Analysis

Studies have discussed the relationship between vegetation and climate under different land use types [70]. Vegetation growth under different land uses depends on certain hydrothermal conditions. Therefore, the effects of temperature and precipitation on cropland, woodland, grassland and built-up land have different effects on vegetation [71–79].

In our study, it was found that the precipitation in the whole study period was insignificantly correlated with the GSN. For different land use changes, we found that there was insignificant negative correlation in the land use change related to woodland and almost insignificant positive correlation in the land use change related to human influence. It indicates that the effect of precipitation on vegetation is affected by the degree of human disturbance. Throughout the study period, the relationship between temperature and GSN was predominantly significant positive among all four land-use types. The proportion of significantly positively correlated areas in natural land use such as woodland and grassland is much greater than that of human-affected land use such as cropland and built-up land. Although the increase of temperature can promote the growth of vegetation, the cropland in the study area is mainly man-made paddy field and irrigation makes the effects of temperature smaller than woodland and grassland. For urban land, there are heat island effect and impervious layer and thus the increase of temperature will increase the evaporation and inhibit the growth of vegetation [79]. For grassland research, the increase in temperature is conducive to the growth of grassland [80].

It was found that the relationship between precipitation and vegetation in the four types of land use was insignificant negative before abrupt change. However, after then, due to the increase of temperature, the acceleration of water evaporation, which caused the positive correlation increase greatly. Among them, the significant positive correlation increased mostly in natural land as woodland and grassland, while the insignificant positive correlation increased mostly in cropland and built-up land that affected by humans. After abrupt change, the correlation shifted from insignificant negative to positive in different land-use changes. However, due to the influence of urbanization, there was mainly insignificant positive correlation in the land use change relevant to built-up land, while others were significantly positively correlated. In the meantime, we found that precipitation in the most changed land had a significant effect on the growth of vegetation compared to unchanged land. The relationship between temperature and vegetation in the four land use types was insignificantly positively correlated whether before or after abrupt change. However, after abrupt change, the insignificance increased in all types of unchanged and changed land use. Among them, due to the increase of temperature, the evaporation of water increased and the water stress was strengthened, which caused the positive correlation between temperature and vegetation to insignificantly negative. However, for woodland, the soil is rich in water, which will weaken the inhibition on vegetation growth to a certain extent, so there were a large number of significant positive to insignificant positive. At the same time, due to the

existence of human-controlled factors in built-up land, the significant positive correlation in the land use types other than built-up land was greatly reduced. Meantime, influenced by urbanization, there was a significant reduction in the positive correlation in the land use changes which they participated.

## 5. Conclusion

Based on the Mann-Kendall test and correlation analysis methods, after abrupt change, the shifts in the vegetation trends and the possible relationships of vegetation dynamics with climate and land use changes have been studied in the Yangtze River Delta during 1982–2016.

Vegetation trends in the Yangtze River Delta changed suddenly in 2000. Although the vegetation was mainly increased significantly from 1982 to 2016, it was mainly insignificant either before or after abrupt change. After abrupt change, the significant increase in GSN had increased greatly in the southern, while the significant decrease increased in the eastern. The decreasing trend of GSN in the cropland, built-up land and the land converted to built-up land increased, while the vegetation with a significant increase in woodland, grassland and most varied land use change types increased. Expect the land affected by urbanization, the vegetation growth under changed land was better than unchanged land.

Throughout the whole study period, the correlation between precipitation and GSN was mainly insignificant positive, except insignificant negative in woodland. After abrupt change, the correlations in the four land use types shifted from insignificantly negative to positive. The significant positive correlation increased mostly in woodland and grassland, while the insignificant positive correlation increased mostly in cropland and built-up land. The proportion of significant positive correlations in most of the changing land was greater than that of unchanging land.

The correlation between temperature and GSN was mainly significant positive throughout the period, however, whether before or after abrupt change, the relationship was mainly insignificant positive. After abrupt change, significant positive correlation decreased and insignificant negative correlation increased. However, for woodland, there was also an increase in insignificant positive correlation. The proportion of significant positive correlations was greater than the conversion to land use affected by urbanization.

Therefore, we can see the necessity of abrupt change analysis to study vegetation dynamics and to assess the effects of climate and land use changes. Our research can provide ecological protection for the sustainable development of the Yangtze River Delta economy.

**Author Contributions:** Conceptualization, H.L.; Data curation, L.W.; Formal analysis, L.W.; Funding acquisition, H.L.; Investigation, Y.R.; Methodology, L.W and H.G.; Project administration, H.L.; Resources, Y.R.; Software, Y.R.; Supervision, H.L.; Visualization, Y.R. and H.G.; Writing—original draft, L.W; Writing—review & editing, H.L and L.W. All authors have read and agreed to the published version of the manuscript.

**Funding:** This research has been supported by National Natural Science Foundation of China (No. 41971382, 31470519) and funded by the Priority Academic Program Development of Jiangsu Higher Education Institutions (164320H116).

**Conflicts of Interest:** The authors declare no conflict of interest.

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
