# Peer review of "Effects of Climate and Land Use changes on Vegetation Dynamics in the Yangtze River Delta, China Based on Abrupt Change Analysis"

_sustainability, doi:10.3390/su12051955_

Round 1
Reviewer 1 Report
The paper entitled “Effects of climate and land use changes on vegetation dynamics in the Yangtze River delta, China based on abrupt change analysis” presents an assessment of the relations between vegetation, land use and climate changes in the Yangtze River delta.
This is a re-submitted manuscript therefore my evaluation regards the entire manuscript keeping an eye to the work that authors did in this new version.
I still insist that the argument is of great interest since future sustainable policies strongly depends on these kind of evaluations.
English is better, but certain phrases remains odd to read for logic sentencing, therefore I suggest to have a deep reading of the work (having attention to spacing and formatting the text); Abstract: too limited your methodological explanation while the 80% is dedicated to results. Re-formulate. Here I have my main doubts. First, the correlation between NDVI, Land Use and Climate is a quite hard issue to deal with, so it is better to make test in little areas, with few, reliable data and demonstrating solid results. This suggestion has not been considered. Second, the dataset. How can you go with an in-depth argumentation if you are using a LULC with 5 classes while analysing LUC and NDVI at 5 km pixel cell??? Are you aware that similar studies are conducted at 5meter cell with more than 60 LULC classes? Third: you spend a lot of rows to describe your data and “what” you have done. Ok but “how” you reached your tables? What about LUC? You cross-tabulation analysis has been an output of a GIS procedure? If so, which one? Please describe in detail. Fourth: why having all three data (climate, LUC and NDVI) you didn’t perform a principal component analysis with a correlation matrix between the components? You mainly describe results here, not “discussing” the things. Moreover, why to focus on insignificant correlations (both positive or negative are insignificant!)See my detailed comments on the attached file.

Author Response
The paper entitled “Effects of climate and land use changes on vegetation dynamics in the Yangtze River delta, China based on abrupt change analysis” presents an assessment of the relations between vegetation, land use and climate changes in the Yangtze River delta.
This is a re-submitted manuscript therefore my evaluation regards the entire manuscript keeping an eye to the work that authors did in this new version.
I still insist that the argument is of great interest since future sustainable policies strongly depends on these kind of evaluations.
Response:
Thanks for your support.
English is better, but certain phrases remains odd to read for logic sentencing, therefore I suggest to have a deep reading of the work (having attention to spacing and formatting the text);
Response:
Thank you very much for your suggestion, we have corrected the logic errors in the manuscript.
Abstract: too limited your methodological explanation while the 80% is dedicated to results. Re-formulate. Here I have my main doubts.
Response:
Thank for your helpful suggestion. In the abstract, we provided more detailed explanation of the methodology as the following (Please see Page1 Line17-21):
Based on Mann-Kendall trend and abrupt change analysis, we monitored vegetation dynamics and its abrupt change in the Yangtze River delta during 1982-2016. With the correlation analysis, we revealed the relationship of vegetation dynamics with climate changes, and with land use changes analysis we studied the effects of land use changes on their relationship.
First, the correlation between NDVI, Land Use and Climate is a quite hard issue to deal with, so it is better to make test in little areas, with few, reliable data and demonstrating solid results. This suggestion has not been considered.
Response:
In fact, we have given the suggestion some serious thought, and we thought it is not
necessary to make test in little areas based on the following considerations:
Our study is to explore the relationships of climate and land use changes on vegetation dynamics at a large scale through the application of remote sensing images. We processed the study pixel-by-pixel, and thus it can reveal the spatial heterogeneity of the impact of climate and land use changes on vegetation dynamics (Please see Page2 Line96-98). In our study, the correlations between NDVI, Land Use and Climate on each pixel were analyzed. Therefore, any area in our study region, whether large or small, were studied, no additional tests are needed for little areas. This kind of pixel-based research is applicable to any scale from local to global. The same similar research is as the following (Please see Page2 Line59-60):
Vegetation dynamics is thought to be driven by both climate and land-use changes at regional, national and even global scales [19-21].
So it can be seen that the study is also reliable in a wide range.
In addition, all our data as introduced by section 2.2 (Data sources) are from official or general data, so the data is reliable!
Thank you!
Reference:
19. Liu, Y.; Li, Y.; Li, S.; Motesharrei, S. Spatial and Temporal Patterns of Global NDVI Trends: Correlations with Climate and Human Factors. Remote Sens. 2015, 7, 13233-13250.
20. Li, S.; Sun, Z.; Tan, M.; Li, X. Effects of rural–urban migration on vegetation greenness in fragile areas: A case study of Inner Mongolia in China. J. Geogr. Sci. 2016, 26, 313-324.
21. Piao, S.L.; Fang, J.Y. Seasonal Changes in Vegetation Activity in Response to Climate Changes in China between 1982 and 1999.Acta Geographica Sinica. 2003, 58, 119-125.
Second, the dataset. How can you go with an in-depth argumentation if you are using a LULC with 5 classes while analysing LUC and NDVI at 5 km pixel cell??? Are you aware that similar studies are conducted at 5meter cell with more than 60 LULC classes?
Response:
Although there are finer scales for land use than 5km such as 5 meter, for NDVI, a spatial resolution of 5km is the highest for the longtime series during 1982-2016, to the best of my knowledge. Our study is to reveal the effects of land use changes on vegetation dynamics during a long time period, so, we have to use land use classification with 5km spatial resolution to match NDVI. Moreover, our study area is the entire Yangtze River Delta, with a very large spatial scope. The data at 5 meters spatial resolution with more than 60 LULC classes in our study area is hard to get, not to mention of data before 2000. Furthermore, analysis of land use changes with so many classifications is very complicated, it is not conducive to our analysis. On the contrary, the classification we used now is widely used in studying the effects of land use on vegetation [Reference].
Thank you!
Reference:
Xin, Z; Xu, J.; Zheng, W. Spatiotemporal variations of vegetation cover on the Chinese Loess Plateau (1981-2006): Impacts of climate changes and human activities.Sci. China Earth Sci. 2008, 51, 67-78.
Third: you spend a lot of rows to describe your data and “what” you have done. Ok but “how” you reached your tables? What about LUC? You cross-tabulation analysis has been an output of a GIS procedure? If so, which one? Please describe in detail.
Response:
The description of the land use transfer matrix has described in detail as the following (Please see Page4 Line158-165):
To perform land use change analysis, firstly, we set cropland, woodland, grassland, build-up land as 1, 2, 3 and 5 respectively, and converted the property value of the land use data in 1990 and 2015 from text to number. Secondly, with the tool of Raster Calculator in ArcGIS 10.2, we used land use data in 1990*10 plus the data in 2015, and then get the changes of land use from 1990 to 2015. For example, 12, 13, 15 means the conversion from cropland to woodland, grassland, and build-up. Finally, we counted the area of each type of land use changes including changed and unchanged in the Attribute Table and put them into the table of transfer matrix.
Thank you!
Fourth: why having all three data (climate, LUC and NDVI) you didn’t perform a principal component analysis with a correlation matrix between the components? You mainly describe results here, not “discussing” the things. Moreover, why to focus on insignificant correlations (both positive or negative are insignificant!)
Response:
Because our research focuses on revealing separately the impacts of climate and land use on vegetation dynamics, principal component analysis is not necessary. We performed our analysis on each pixel, so principal component analysis is not appropriate. We can consider it in our future study.
Insignificant correlations between precipitation and vegetation dynamics during the whole period were analyzed to compare the correlations with those before or after abrupt change, and further to emphasize the necessity of abrupt change analysis.
Thanks.
1. Line 20 you cannot open-up with results at the thir raw of your abstract... 1 research question and introduction, 2 methodology, 3 results 4 main innovations. (equally distributed with a more emphasys to the method)
Response:
Thank you for your suggestions, we have added more description of the method as the following (Please see Page1 Line17-21):
Based on Mann-Kendall trend and abrupt change analysis, we monitored vegetation dynamics and its abrupt change in the Yangtze River delta during 1982-2016. With the correlation analysis, we revealed the relationship of vegetation dynamics with climate changes, and with land use changes analysis we studied the effects of land use changes on their relationship.
2. Line 21 growing season NDVI does not means anything since NDVI is an index... maybe the growing of NDVI is correct...
Response:
The growing season means the period of the year that is warm enough for plants to grow. The growing season NDVI in our study means the average NDVI during the growing season. We changed “growing season NDVI” to “NDVI during growing season” (Please see Page1 Line21-22). And we explained the growing season in more detail as the following (Please see Page1 Line21-22):
the NDVI during growing season that is represented as GSN.
Thank you!
3. Line 69 see logic sentencing...
Response:
Thank you for your suggestion and we changed as the following (Please see Page2 Line69-70):
Human activities such as urbanization, mining, and reforestation have changed land use types greatly and further caused corresponding degradation or greening of vegetation [28-30].
Reference:
28. Li, S.; Sun, Z.; Tan, M.; Li, X.Effects of rural-urban migration on vegetation greenness in fragile areas: A case study of Inner Mongolia in China. J. Geogr. Sci. 2016, 26, 313-324.
29. Wang, J.; Wang, P.; Qin, Q.; Wang, H. The effects of land subsidence and rehabilitation on soil hydraulic properties in a mining area in the Loess Plateau of China. Catena. 2017. 159, 51–59.
30. Zipper, C.E.; Burger, J.A.; Skousen, J.G.; Angel, P.N.; Barton, C.D.; Davis, V.; Franklin, J.A. Restoring Forestsand Associated Ecosystem Services on Appalachian Coal Surface Mines. Environ. Manag. 2011, 47, 751–765.
4. Line 78 in this introduction you are mixing the climate change and the land use change as phenomena that affect vegetation... which is true but it is better to clarify the two phenomena separately: climate change is somehow an exogenous phenomena, while LUC is endogenous... if you merge both are you sure you can determine a cause-effect mechanism?
Response:
In fact, we have discussed separately the effects of climate change and land use changes as the following:
First, we discussed the relationship between land use change and vegetation dynamics in Section 3.1.2. Then we discussed the relationship between vegetation dynamics and climate change in Section 3.2, and finally discussed the relationship between vegetation dynamics and climate change under different land use changes
Thank you!
5. Line 96 what does it means based on the pixel scale? it means you used raster datasets? with what grid cell dimension? why raster an not vector and why don't refer to a scale of analysis?
Response:
Yes, we used raster datasets for analysis. The pixel scale means that we have performed our analysis on each pixel, such as correlation analysis and land use change analysis. Unlike
vector, raster can be converted into a data matrix in the calculation, it is more convenient to perform scientific calculations for each pixel than vector. We described the grid size as the following:
(Please see Page4 Line125-128) In this study, the NDVI dataset was downloaded from the University of Arizona website (https://vip.arizona.edu) during 1982-2016 with a time resolution of 30 days and a spatial resolution of 5 km, which is one of the longest time series [48]. Then, all the climate and land use data were resampled to match the NDVI data with the nearest neighbor approach.
Thank you!
Reference:
48. Qu, S.; Wang, L.; Lin, A.; Zhu, H.; Yuan, M. What drives the vegetation restoration in Yangtze River basin, China: Climate change or anthropogenic factors? Ecol Indic. 2018, 90, 438-450.
6. Line 99 abrupt change of what?
Response:
It means abrupt change of vegetation. We changed it as the following:
We changed “after abrupt change” to “before or after vegetation abrupt change” (Please see Page3 Line100).
Thanks.
7. Line 101 at the pixel level?
Response:
Yes.
Thank you!
8. Line 102 this is quite complicated... three relations inside...
Response:
Based on the land use transfer matrix, the relationship between vegetation and climate under different land use changes is analyzed in our study. And such research (Li, J et,al. 2004), they analyzed the relationship between rainfall and NDVI pixel by pixel, and discussed this relationship on different land use performances for Senegal.
Reference:
Li, J.; Lewis, J.; Rowland, J.; Tappan, G.; Tieszen, L.L. Evaluation of land performance in Senegal using multi - temporal NDVI and rainfall series. Arid En-viron. 2004, 59, 463―480.
9. Line 104 what? (the verb... we studied, we explored...)
Response:
We added “we studied” in the study (Please see Page3 Line105).
Thank you!
10. Line 134 This explanation of VIP is out of this work... why are you sponsoring the VIP?
Response:
We use the description of VIP to explain the scientific nature of the data. And they have high spatial resolution in long-term sequence of NDVI data. At the same time we removed redundant descriptions as the following (Please see Page4 Line128-132):
Our NDVI data comes from the Vegetation Index and Phenology (VIP) Laboratory, which specializes in designing highly specialized data processing science algorithms and in generating consistent, high quality and well characterized Climate Data Record (CDR) and Earth Science Data Record (ESDR) long term single- and multi-satellite data products in support of global change studies.( https://vip.arizona.edu)
Thanks.
11. Line 135 why exploring only growing... and not all the trend? growth and de-growth?
Response:
We studied NDVI during the growing season (from March to October), and then we studied all trends of GSN (NDVI during the growing season) through the MK method, including increasing, decreasing and insignificant changes
Thank you!
12. Line 141 what is DEM?
Response:
It means Digital Elevation Model. And we added “elevation (DEM)” (Please see Page4 Line140).
Thanks.
13. Line 141 these means meterological?
Response:
Yes. These are climate datasets (monthly temperature and precipitation data) for 58 meteorological stations in the study area. And we changed “these” to “meteorological” as the following (Please see Page4 Line141):
meteorological data (monthly temperature and precipitation data)
Thank you!
14. Line 142 what kind of?
Response:
We added “with a 5 km spatial resolution to match the NDVI dataset” in the study (Please see Page4 Line141).
Thank you!
15. Line 148 think here you have to explain beter since this influences a lot your analysis.. why not using kridging for example?
Response:
Because this method has more advantages when dealing with meteorological interpolation, it can consider covariates of latitude, longitude and elevation (DEM), which kriging interpolation cannot. We described as the following:
(Please see Page4 Line140-143):
Based on the covariates of latitude, longitude and elevation (DEM) of each meteorological site (from the US Geological Survey), meteorological data (monthly temperature and precipitation data) were interpolated and resampled to continuous surface data with a 5 km spatial resolution to match the NDVI dataset, using thin-plate smoothing spline model in ANUSPLIN [49].
(Please see Page4 Line146-147):
This method can not only interpolate the independent variables, but also introduce a linear covariate submodel which provides the possibility of introducing multiple influence factors.
Thanks.
Reference:
49. Hijmans, R.J.; Cameron, S.E.; Parra, J.L.; Jones, P.G.; Jarvis, A.J. Very high resolution interpolated climate surfaces for global land areas. Int. J. Climatol. 2010, 25, 1965–1978.
16. Line 160 how you employed you land use change analysis? which method?
Response:
We described it in more detail as the following (Please see Page4 Line158-168):
To perform land use change analysis, firstly, we set cropland, woodland, grassland, build-up land as 1, 2, 3 and 5 respectively, and converted the property value of the land use data in 1990 and 2015 from text to number. Secondly, with the tool of Raster Calculator in ArcGIS 10.2, we used land use data in 1990*10 plus the data in 2015, and then get the changes of land use from 1990 to 2015. For example, 12, 13, 15 means the conversion from cropland to woodland, grassland, and build-up. Finally, we counted the area of each type of land use changes including changed and unchanged in the Attribute Table and put them into the table of transfer matrix. We used the land use change map to superimpose the vegetation trend map to obtain the impact of land use on vegetation dynamics. In addition, the maps of the correlation between climate and vegetation were superimposed with the land use change map to analyze the effects of different land use changes.
Thank you!
17. Line 162 Please consider that 5km grid isn't a good resoltion to perform a study of relation between NDVI, LUC and Climate Change... 5 km can be fine only for climate change (see that actually NDVI and LUC are detected at 5 meters, with a land use classification with more than 60 classes...)
Response:
Although there are finer scales for land use than 5km such as 5 meter, for NDVI, a spatial resolution of 5km is the highest for the longtime series from 1982-2016, to the best of my knowledge. Our study is to reveal the effects of land use changes on vegetation dynamics during a long time period, so, we have to use land use classification with 5km spatial resolution to match NDVI. Moreover, our study area is the entire Yangtze River Delta, with a very large spatial scope. The data at 5 meters spatial resolution with more than 60 LULC classes in our study area is hard to get, not to mention of data before 2000. Furthermore, analysis of land use changes with so many classifications is very complicated, it is not conducive to our analysis. On the contrary, the classification we used now is widely used in
studying the effects of land use on vegetation [Reference].
Thanks.
Reference:
Xin, Z; Xu, J.; Zheng, W. Spatiotemporal variations of vegetation cover on the Chinese Loess Plateau (1981-2006): Impacts of climate changes and human activities.Sci. China Earth Sci. 2008, 51, 67-78.
18. Line 164 but why you need to use this test! it's not well explained: if you have to detect NDVI changes you can see how a pixel changed between temporal thresholds...
Response:
Because we studied the trend of NDVI changes, not the changes of NDVI value. At the same time, the advantage of the Mann-Kendall trend test is that the sample does not need to follow a specific distribution, and is rarely disturbed by outliers, and the calculation is simple (Please see Page4 Line171-173).
Thank you!
19. Line 214 abrupt change of what? climate?
Response:
To avoid ambiguity, we added “vegetation” in the study (Please see Page6 Line217).
Thank you!
20. Line 249 ok but can you explain in detail how did you calculated your matrix?
Response:
We described this in Data source as the following (Please see Page4 Line159-166):
To perform land use change analysis, firstly, we set cropland, woodland, grassland, build-up land as 1, 2, 3 and 5 respectively, and converted the property value of the land use data in 1990 and 2015 from text to number. Secondly, with the tool of Raster Calculator in ArcGIS 10.2, we used land use data in 1990*10 plus the data in 2015, and then get the changes of land use from 1990 to 2015. For example, 12, 13, 15 means the conversion from cropland to woodland, grassland, and build-up. Finally, we counted the area of each type of land use changes including changed and unchanged in the Attribute Table and put them into the table of transfer matrix.
Thanks.
21. Line 259 see spacing
Response:
We added the spacing (Please see Page8 Line266).
Thanks.
22. Line 378 meaning that from 1982 to 2016 grassland and woodland increased?
Response:
For this sentence, it means that more woodland and grassland showed significant increasing trends than cropland and built-up land from 1982 to 2016. To avoid ambiguity, we modified this as the following (Please see Page14 Line381-383):
In particular, our study found that the area showing significant increasing trend in natural land use types such as woodland and grassland is much larger than that of human-affected land use types such as cropland and built-up land.
Thank you!
23. Line 388 NDVI and Land Use are strongly correlataed. It's odd to think that NDVI grows where antropic areas de-growth...
Response:
We are very sorry that the south we describe is the south of the Yangtze River Delta. To avoid ambiguity, we have modified it as the following (Please see Page14 Line390-391):
Some studies have suggested that the increase in the vegetation of woodland in the south of the Yangtze River Delta was related to human-related policies such as ecological protection projects [65].
Thanks.
Reference:
65. Jiang, Q.; Cheng, Y.; Jin, Q.; Deng, X.; Qi, Y. Simulation of Forestland Dynamics in a Typical Deforestationand Afforestation Area under Climate Scenarios. Energies. 2015, 8, 10558-10583.
24. Line 441 But why you are discussing what is "insignificant"? Insignifican means anything...
Response:
We discussed this are in order to compare the relationship between precipitation and GSN during the whole period and before or after abrupt change, and emphasize the necessity of abrupt change analysis. However, with your suggestions, we have simplified this part as the following (Please see Page15 Line445-446):
In our study, it was found that the precipitation in the whole study period was insignificantly correlated with the GSN.
Thank you!
25. Line 454 all this specific considerations are out of your analysis since you have any interal differentiation: in urban areas and in croplands
Response:
We have rewritten and reanalyzed it as the following (Please see Page15 Line451-457):
The proportion of significantly positively correlated areas in natural land use such as woodland and grassland is much greater than that of human-affected land use such as cropland and built-up land. Although the increase of temperature can promote the growth of vegetation, the cropland in the study area is mainly man-made paddy field, and irrigation makes the effects of temperature smaller than forestland and grassland. For urban land, there are heat island effect and impervious layer, and thus the increase of temperature will increase the evaporation and inhibit the growth of vegetation [83].
Thank you!
Reference:
83. Li, L.; Zha, Y. Satellite-Based Spatiotemporal Trends of Canopy Urban Heat Islands and Associated Drivers in China’s 32 Major Cities. Remote Sens. 2019, 11, 102.

Reviewer 2 Report
Congratulations to the authors. This new version of the manuscript have addressed all the concerns of the previous round of reviews. Please, check the following mistakes in the new version.
Line 101: check space between "the" and "spaces".
Line 154: 1st
Author Response
Reviewer2.
Congratulations to the authors. This new version of the manuscript have addressed all the concerns of the previous round of reviews. Please, check the following mistakes in the new version.
Response:
Thank you very much for your approval.
1. Line 101: check space between "the" and "spaces".
Response:
We added the spacing (Please see Page3 Line102).
Thanks.
2. Line 154: 1st
Response:
We have modified this (Please see Page4 Line154).
Thank you

Round 2
Reviewer 1 Report
Here my comments:
- Abstract: too limited your methodological explanation while the 80% is dedicated to results. Re-formulate.
- Delete repetions and simplyfy certain phrases (in particular the terms change and changes in the introduction).
- divide the description of data source to data processing.
Good luck.
Author Response
Response to Reviewer 1 Comments Abstract: too limited your methodological explanation while the 80% is dedicated to results. Re-formulate. Response: Thank you for your suggestion. We added a description of the method and deleted some results as the following (please see Page 1 Line16-36): Vegetation dynamics is thought to be affected by climate and land use changes. However, how the effects varied after abrupt vegetation changes remains unclear. Based on the Mann-Kendall trend and abrupt change analysis, we monitored vegetation dynamics and its abrupt change in the Yangtze River delta during 1982-2016. With the correlation analysis, we revealed the relationship of vegetation dynamics with climate changes (temperature and precipitation) pixel-by-pixel, and then with land use changes analysis we studied the effects of land use changes (unchanged or changed land use) on their relationship. Results showed that: (1) the NDVI during growing season that is represented as GSN showed an overall increasing trend and had an abrupt change in 2000. After then, the area percentages with decreasing GSN trend increased in cropland and built-up land, mainly located in the eastern, while those with increasing GSN trend increased in woodland and grassland, mainly located in the southern. Changed land use, except the land conversions from/to built-up land, is more favor for vegetation greening than unchanged land use (2) after abrupt change, the significant positive correlation between precipitation and GSN increased in all unchanged land use types, especially for woodland and grassland (natural land use), and changed land use except built-up land conversion. Meanwhile, the insignificant positive correlation between temperature and GSN increased in woodland, while decreased in the cropland and built-up land in the northwest (3) after abrupt change, precipitation became more important and favor, especially for natural land use. However, temperature became less important and favor for all land use types, especially for built-up land. This research indicates that abrupt change analysis will help to effectively monitor vegetation trend and to accurately assess the relationship of vegetation dynamics with climate and land use changes. Delete repetions and simplyfy certain phrases (in particular the terms change and changes in the introduction). Response: We deleted some “changes” and replaced “abrupt change” with some pronouns in the introduction. Moreover, we deleted repeats and simplified certain phrases in other parts of the manuscript. Thanks. divide the description of data source to data processing. Response: We modified it (please see Page 4 Line122). Thank you!
This manuscript is a resubmission of an earlier submission. The following is a list of the peer review reports and author responses from that submission.
Round 1
Reviewer 1 Report
General comments:
The submitted paper is a very interesting approach to the application of abrupt change analysis with Mann-Kendall trend analysis to the assessment of effects of climate changes and land use on vegetation changes in China. The methodology design is appropriate, the statistical analysis is well grounded, and the results are significant and with scientific soundness. However, the paper has critical deficiencies that should be included in future versions, including some lacks in the presentation of the manuscript, which would need extra amendments.
Structure of the manuscript
Title
The title is very proper for the presented study.
Abstract
I found the abstract adequate and concise. I suggest including much more information about the methodology.
I suggest removing the points of lines 20, 26 and 29.
Please, do not use acronyms in the abstract. Write instead “growing season NDVI (GSN)”.
Line 22: check punctuation.
Introduction
The redaction is concise but there are many abrupt changes between paragraphs; please, improve the writing style of this section.
I suggest stating clearly which is the scientific gap of the study. I suggest relocating the sentences of line 54 to 59 to the end of the introduction section.
What do you mean in this sentence (Lines 83-86)? “Therefore, it is of great significance to study the response of vegetation change to land use change and climate change in the Yangtze River Delta, but how the effects changed before and after vegetation abrupt changes remained unclear.”
Lines 45, 62, 64, 66, 68, 69: check punctuation.
Line 91: please include citation for the Mann-Kendall test.
Line 92: check grammar.
Lines 94-95: please, rewrite this sentence, it is too confusing.
Material and methods
Line 101: please check grammar.
Line 102: The study area.
Line 103: Specify which is the growing season.
Line 108: Under which classification of land uses are you working?
Figure 1: I suggest modifying the scale bar, with divisions multiple of 100.
Data source: you work with different spatial resolutions; how do you integrate them?
Line 118: check punctuation.
Line 124: Which is the error of the interpolation process?
Line 127: check punctuation.
Line 132: check punctuation.
For which purpose you use the Mann-Kendall Trend test? Please justify it and write an introduction for this sub-section.
Lines 159 – 160: check grammar.
Results
Line 168: space missing.
Line 170: check grammar.
Figure 3: What does UF and UB mean?
Line 175: space missing.
Lines 184 – 188: check punctuation and spaces.
Table 1: Please, reorder the presentation of the first rows of the 3rd and 4th columns.
Line 198: check punctuation and spaces.
Line 199: Why do you calculate the changes until 2015 and not until 2016, as specified before?
Figure 5: Clearly show the remaining land uses in the legend.
Table 2: check punctuation and spaces (first row, last column).
Line 209: check spaces missing.
Table 3, 4, 6, 7, 9 and 10: check punctuation and spaces at the bottom of the tables.
Lines 232 and 234: please, specify that this percentages display % of the study area.
Lines 232-241: check punctuation and spaces.
Line 242: check grammar.
Table 5: How do you classify the correlation values displayed in the first column? Justify the use of these classes or show the correlation values for each class.
Line 262: improve the redaction of this sentence.
Lines 275 and 276: please, specify that this percentages display % of the study area.
Lines 274 – 286: please, clearly state within the text which are the correlation values.
Discussion
I want to congratulate the authors for the wonderful discussion section, which includes their own results linked to the most current scientific evidence on the subject.
Line 319: Line 209: check spaces missing.
Line 319: Line 242: check grammar.
Conclusions
I suggest including much more information about the methodology used for the study. This conclusions section is only focused in the results.
I suggest removing the points of lines 429 and 437.
Reviewer 2 Report
The paper entitled “Effects of climate changes and land use on vegetation changes in the Yangtze River delta, China based on abrupt change analysis” presents an assessment of the relations between vegetation, land use and climate changes in the Yangtze River delta.
The argument is of great interest since future sustainable policies strongly depends on these kind of evaluations.
Nonetheless, this manuscript has several flaws and weaknesses.
To what concern the style the mayor problem is the English sentencing, which is not accurate. Bad, long sentences often neglect the possibility to understand the meaning of the text. Punctuation has many mistakes, spacing is often missing, many repetitions are founded (see the title for example).
To what concern the structure, this work is ambitious but is lacking in giving to the reader the possibility to fully understand the correctness of the analysis. The correlation between NDVI, Land Use and Climate is a quite hard issue to deal with, so it is better to make test in little areas, with few, reliable data and demonstrating solid results.
Then it comes the method: I cannot understand how NDVI has been calculated. It is an average value in the catchment area? Is an average absolute value? Please consider that NDVI can grow while the surface of un-built land decrease! Are you considering this? So to demonstrate your analysis you should adopt a weighted average value of NDVI (NDVI*land use extension/catchment area)
Second: the land use has been evaluated using 4 land use classes. It’s to poor! How can you get an in-depth relation between vegetation and land use if you are analysing 4 classes? Are you aware of how general is a detection with 4 classes? What was the scale of detection? Consider that the land use classification influences your analysis! And how did you perform your LUC analysis? You didn’t introduce adequately LUC analysis in your text
Third: the climatic part. Understanding the relations between climate data and vegetation require a huge quantity of climatic data in a long time-series… your deductions can be conditioned by local frequencies of events and not properly by the climatic variability.
At least I suggest you to re-write your manuscript, shortening and clearing the methodology, selecting few variables (not including climate for examples) and using accurate data (an accurate land use classification) while describing in a detailed manner everything concerning your analysis and making clear and simple consideration around your assessment.
Good luck!
